# Protective Effect of Cocoa Bean Shell against Intestinal Damage: An Example of Byproduct Valorization

**DOI:** 10.3390/antiox10020280

**Published:** 2021-02-12

**Authors:** Daniela Rossin, Letricia Barbosa-Pereira, Noemi Iaia, Barbara Sottero, Alice Costanza Danzero, Giuseppe Poli, Giuseppe Zeppa, Fiorella Biasi

**Affiliations:** 1Department of Clinical and Biological Sciences, University of Turin, 10043 Orbassano, Italy; d.rossin@unito.it (D.R.); noemi.iaia@unito.it (N.I.); barbara.sottero@unito.it (B.S.); alice.danzero@edu.unito.it (A.C.D.); giuseppe.poli@unito.it (G.P.); 2Department of Analytical Chemistry, Nutrition and Food Science, Faculty of Pharmacy, University of Santiago de Compostela, 15782 Santiago de Compostela, Spain; letricia.barbosa.pereira@usc.es; 3Department of Agricultural, Forestry, and Food Sciences (DISAFA), University of Turin, 10095 Grugliasco, Italy; giuseppe.zeppa@unito.it

**Keywords:** polyphenols, dietary oxysterols, epithelial barrier, intestinal inflammation, IL-8, MCP-1, tight junctions, Nrf2, theobromine

## Abstract

Background: Cocoa bean shell (CBS), a main byproduct of cocoa processing, represents a source of components such as polyphenols and methylxanthines, which have been associated with a reduced risk of several diseases. Therefore, CBS has potential application as a food ingredient. Intestinal mucosa is exposed to immune and inflammatory responses triggered by dietary agents, such as oxysterols, which derive from cholesterol oxidation and are pro-oxidant compounds able to affect intestinal function. We aimed at investigating the capability of the Forastero cultivar CBS, added or not added to ice cream, to protect against the intestinal barrier damage induced by a dietary oxysterol mixture. Methods: Composition and antioxidant capacity of in vitro digested CBS and CBS-enriched ice cream were analyzed by high-performance liquid chromatography and 1,1-diphenyl-2-picryl-hydrazyl radical-scavenging assay, respectively. CaCo-2 cells differentiated into enterocyte-like monolayer were incubated with 60 µM oxysterol mixture in the presence of CBS formulations. Results: The oxysterol mixture induced tight junction impairment, interleukin-8 and monocyte chemoattractant protein-1 cell release, and oxidative stress-related nuclear factor erythroid 2 p45-related factor 2 response Nrf2. Both CBSs protected cells from these adverse effects, probably thanks to their high phenolic content. CBS-enriched ice cream showed the highest antioxidant capacity. Theobromine, which is in high concentrations of CBS, was also tested. Although theobromine exerted no effect on Nrf2 expression, its anti-inflammatory cooperating activity in CBS effect cannot be excluded. Conclusions: Our findings suggest that CBS-enriched ice cream may be effective in the prevention of gut integrity damage associated with oxidative/inflammatory reactions.

## 1. Introduction

Chocolate consumption has recently been on the rise thanks to wider commercial availability around the world, particularly in the European Union and North America, having an annual consumption growth rate of 1.7% and 3.6% annually, respectively [1]. Besides being universally famous as one of the top comfort foods, chocolate is now well known for its healthy effects, primarily on the cardiovascular system [2] and as an antidepressant [3]. In addition, its potential benefits may play an important role in the prevention of other pathological conditions, including cancer [4,5,6,7].

Cocoa beans, the seeds of the tropical *Theobroma cacao* L. tree, are the source of the main ingredient used for chocolate and cocoa-derivative food production, namely cocoa mass (liquor). To obtain the cocoa mass, cocoa beans undergo several treatments, among which are roasting and husking, which are necessary to remove the so-called cocoa bean shell (CBS), a fine shell covering the bean [8].

CBS accounts for 12–20% of bean weight [8] and represents an important economic issue in the cocoa industry for its disposal [1,9]. To minimize waste production and relative costs, alternative innovative strategies for CBS recycling as possible additives (e.g., supplements, integrators) in functional food and beverages are now being evaluated [8,10].

Recent studies have, in fact, recognized CBS as a source of disease-preventing components, including fibers, minerals, methylxanthines (caffeine, theobromine), and polyphenols (catechins, epicatechins, procyanidins), whose actual content highly depends on factors such as cocoa genetic variety, cultivation, and cocoa bean processing [8,9,10,11,12].

Health-promoting effects of polyphenols and methylxanthines have been widely shown. Polyphenols, in particular, show free radical scavenging and metal chelating effects, and are recognized as strong modulators of redox sensitive-pathways which are capable of counteracting cell and tissue oxidative damage in several chronic human pathologies, including inflammatory gut diseases and colorectal cancer [13,14,15].

Methylxanthines are mainly known for their smooth muscle relaxant, diuretic, and coronary vasodilator properties. The main action mechanisms of caffeine and theobromine are adenosine receptors’ blockade and phosphodiesterase inhibition, which seem to account for their anti-inflammatory and antitumoral activity [16,17].

Besides that, theobromine exhibits other important adenosine receptor-independent effects, such as the reduction of cellular oxidative stress or regulation of gene expression [16,17].

Polyphenols, and very recently also theobromine, have been proposed as potential modulators of intestinal permeability [18,19], whose perturbation is an important feature of gut function [20].

Intestinal mucosa is indeed particularly exposed to environmental and dietary agents responsible for sustaining immune and inflammatory responses [20]. The activation of these processes can trigger an overproduction of oxidant species, participating in intestinal epithelial permeability damage and mucosal tight junction (TJ) derangement [21,22].

A dietary regimen rich in fruit and vegetables and/or the intake of supplements is able to reduce inflammation and preserve intestinal barrier integrity [23].

On the contrary, the Western-style diet represents one of the main risks for altered intestinal integrity [24]. The dietary animal fats that can undergo oxidation during food processing have been suggested to exert inflammatory and oxidative insults in gut mucosa [25]. The oxidative cholesterol derivatives, oxysterols, are involved in modulating inflammation, fibrosis, and cell death, thus playing a role in the pathogenesis of several chronic diseases, such as inflammation-related bowel diseases [26].

We previously proved that a mixture of oxysterols corresponding to high dietary consumption of cholesterol triggered oxidative reactions and inflammatory responses in human enterocyte-like CaCo-2 cells. Nicotinamide adenine dinucleotide phosphate (NADPH) oxidase, mitogen-activated protein (MAP) kinase, and nuclear factor kappa-light-chain-enhancer of activated B cells (NF-kB) appeared to be the main signaling pathway involved [27]. Those events were prevented in the same cellular model by red wine extracts showing high phenolic content [27].

Oxysterols have also been found to modulate the expression of genes controlled by stress-induced nuclear factor erythroid 2 p45-related factor 2 (Nrf2) depending on the type of target cells, the environmental conditions, and the concentration and type of oxysterols [28]. Nrf2 is a key transcription factor whose activation represents a protecting adaptive cell response against redox stressors by promoting codification for various stress-related detoxifying and antioxidant enzymes [29]. A recent study showed that dietary oxysterols were able to induce inflammation in CaCo-2 cells by also activating the immune system-related Toll-like Receptor (TLR) superfamily members TLR2 and TLR4 [30]. In those experiments, CBS extracts from Honduras cocoa beans were effective in protecting intestinal mucosa from oxidative stress and inflammation through TLR downregulation [30].

The aim of our current investigation is to give in vitro further evidence of CBS positive effect against oxysterol-induced oxidative and inflammatory intestinal barrier damage.

Although increasing literature suggest several healthy properties for polyphenols and methylxanthines significantly present in CBS, most of the evidence about antioxidant and anti-inflammatory activities are based on studies conducted on single compounds, while the bioactivity of CBS, and subsequently, its use in functional foods has not been in-depth investigated yet.

We think that our findings on the beneficial effects of CBS-enriched ice cream give value to cocoa byproducts as healthy food recycled supplements for the prevention of inflammatory gut diseases.

## 2. Materials and Methods

### 2.1. Materials

Chemicals for the analyses of phenolic compounds, methylxanthines, and antioxidant activity: Ethanol (≥99.9%), formic acid (98–100%), 6-hydroxy-2,5,7,8-tetramethylchroman-2-carboxylic acid (97%) (Trolox), 2,2-diphenyl-1- picrylhydrazyl (95%) (DPPH), Folin-Ciocalteu’s phenol reagent, Na2CO3 (≥99.5%), NaNO2 (≥99%), AlCl3 (≥99%), vanillin (≥99%), HCl, (+)-catechin hydrate (> 98%), quercetin-3-β-D-glucoside (≥90%), theobromine (≥98.5%), and caffeine (≥98.5%), obtained from Sigma-Aldrich (Milan, Italy). Gallic acid and (−)-epicatechin (≥99%) were from Fluka (Milan, Italy).

Chemicals for the simulated gastrointestinal digestion: Na2CO3 (≥99.5%) and HCl were obtained from Sigma-Aldrich (Milan, Italy). NaHCO3, NaCl, KCl, K2HPO4, KH2PO4, MgCl2·6H2O, (NH4)2CO3, NaOH, and CaCl2·2H2O were provided by Carlo Erba (Milan, Italy). The enzyme α-amylase from *Bacillus* sp., pepsin from porcine gastric mucosa, pancreatin from porcine pancreas, and bile salts were supplied by Sigma-Aldrich (Milan, Italy).

Chemicals and materials for cell culture and treatment: Cell culture glucose-enriched Dulbecco’s modified Eagle’s medium (DMEM), fetal bovine serum (FBS), and trypsin (5 g/L solution) were from Euroclone SpA (Milan, Italy). 5α,6α-Epoxycholesterol (α-epox), 5β, 6β-epoxycholesterol (β-epox), 7-ketocholesterol (7K), 7α-hydroxycholesterol (7α-HC), and 7β-hydroxycholesterol (7β-HC) were purchased from Avanti Polar Lipids (Alabaster, AL, USA). Multiwell plates and 13 mm glass slides were from VWR International Srl (Milan, Italy).

Materials and kits for the biomolecular analyses: Enzyme-linked immunosorbent assay (ELISA) kits for the evaluation of human interleukin (IL)-8 and monocyte chemoattractant protein (MCP)-1 were obtained from PeProtech (DBA Italia Srl, Segrate, Milan, Italy). Santa Cruz Biotechnology (DBA Italia Srl, Segrate, Milan, Italy) provided the polyclonal primary antibodies rabbit anti-junctional adhesion molecule-A (JAM-A) (SC-25629), mouse anti-occludin (SC-133256), and mouse anti-claudin 1 (SC-166338). Cell Signaling Technology (Euroclone SpA, Milan, Italy) provided the secondary antibodies anti-rabbit IgG HRP-conjugated (7074S) and anti-mouse IgG HRP-conjugated (7076S). Thermo Fisher Scientific (Life Technologies Italia, Monza, Italy) provided goat anti-rabbit IgG-Alexa Fluor 546 and goat anti-mouse IgG-Alexa Fluor 488, lithium dodecyl sulfate (LDS) Sample Buffer 4X, and dithiotreitol (DTT) Sample Reducer 10X. Proteases inhibitors cocktail “cOmplete ULTRA Tablets Mini EASYpack” and the nicotinamide adenine dinucleotide (NADH) were purchased by Roche SpA (Monza, Italy). 

Bio-Rad protein assay dye reagent and ECL^®^ Western Blotting System were from Bio-Rad Srl (SIAL, Rome, Italy). Hybond ECL nitrocellulose membrane was from GE Healthcare Srl (Milan, Italy).

TRIzol reagent was from Invitrogen (S. Giuliano Milanese, Italy). High-Capacity Complementary DNA (cDNA) reverse transcription kit, TaqMan gene expression assay kits for human Nrf2 and β-actin, TaqMan Fast Universal PCR (polymerase chain reaction) master mix, and TaqMan Array 96-well plates were from Applied Biosystems (Monza, Italy). Milli-Q filter system ultrapure water was from Millipore (Milan, Italy). All other not specified reagents and chemicals were obtained from Sigma-Aldrich (Milan, Italy).

### 2.2. Cocoa Bean Shell Sample Preparation and Ice Cream Formulation

Pastiglie Leone Srl (Turin, Italy) kindly supplied cocoa bean shell (CBS) samples yielded from cocoa beans of Forastero cultivar from Sao Tomé.

CBS samples were reduced to fine particles in a ball mill to obtain a powder, 95% of which showed a size lower than 20 µm (measured by a Mastersizer 3000, Malvern Instruments Limited, Worcestershire, UK).

CBS samples were then used as fat replacers in the preparation of ice cream (IC) at different concentrations (0, 2, 4, 6 and 8 wt.% of CBS) [31]. The ice cream obtained by 4% fortified CBS flour (CBS-IC) was selected because of its reduced fat and increased fiber contents compared with plain IC (considered as control sample). CBS-IC formulation maintained similar structural characteristics to the chocolate ice cream (control) and higher acceptance after the consumer test. The formulations used for ice cream production are shown in Appendix A. The chemical and nutritional characterization of samples are shown in Appendix A.

CBS and CBS-IC were used for experiments with the intestinal cells after in vitro digestion as hereafter reported.

### 2.3. In Vitro Simulated Gastrointestinal Digestion

The simulated gastrointestinal digestion related to the oral, gastric, and small intestinal tracts was performed to mimic human digestion according to the routine standardized static in vitro INFOGEST protocol suitable for food described by Minekus and colleagues [32]. For each digestion phase, digestive juices were prepared in Milli-Q water every 2 days for the mouth (Simulated Saliva Fluid, SSF), stomach (Simulated Gastric Fluid, SGF), and small intestine (Simulated Duodenal Fluid, SDF), following protocol indications as described by the authors. Each digestion fluid had a different electrolyte composition, and the pH was adjusted to 7 in SSF, to 3 in SGF, and 7 in SIF using HCl or NaOH 1M.

For the oral phase, 5 g of ice cream were mixed with 5 mL SSF containing human salivary α-amylase solution (1500 U/mL). Then, the mixture was adjusted to pH 7 ± 0.2 and incubated at 37 °C in a shaking water bath for 2 min.

For the gastric phase, 10 mL SGF containing porcine pepsin (2000 U/mL) were added to the ice cream. The mixture was then adjusted to pH 3 ± 0.2 and incubated at 37 °C for 2 h.

Finally, 20 mL of SDF containing pancreatin (100 U/mL trypsin activity) and porcine bile extract (10 mM) were added in the last step of digestion. The final mixture was adjusted to pH 7 ± 0.2 and samples were incubated with orbital agitation at 37 °C for further 2 h. After the complete digestion, pH was adjusted to 5.4, and the samples were immediately kept in ice to minimize enzyme activity and then centrifuged at 12,500× *g* at 4 °C for 10 min. Based on the bioaccessibility studies performed in our laboratory [33], the supernatants were filtered through a 0.22 μm cellulose acetate membrane filter (VWR, Milan, Italy) and stored at −20 °C for further analyses. The simulated gastrointestinal digestion was performed in triplicate for each sample. CBS flour was also digested without IC to evaluate the effect of food matrix on bioactive compound bioavailability.

### 2.4. Reverse Phase-High-Performance Liquid Chromatography-Photodiode Array Analysis of Cocoa Bean Shells and Ice Cream Compounds

CBS and IC chromatographic analyses before and after completed digestion were performed with reverse phase-high-performance liquid chromatography-photodiode array (RP-HPLC-PDA) Thermo-Finnigan Spectra System (Thermo-Finnigan, Waltham, USA) equipped with a P2000 binary gradient pump, SCM 1000 degasser, AS 3000 automatic injector and Finnigan Surveyor PDA Plus detector.

The phenolic compounds and methylxanthines were separated at 35 °C on a reverse phase Kinetex Phenyl-Hexyl C18 column (150 × 4.6 mm internal diameter and 5 μm particle size) (Phenomenex, Castel Maggiore, Italy). The mobile phase consisted of 0.1% formic acid (solvent A) and 100% methanol (solvent B). A linear gradient elution method was applied as follows: 0–2 min with 90% A and 10% B; 2–18 min from 10 to 50% B; 18–40 min, from 50 to 80% B; 40–42 min, up to 90% B. Column re-equilibration was then performed by 90% A and 10% B 42–45 min elution. The mobile phase flow rate was 1.0 mL/min, and the sample injection volume was 10 μL. ChromQuest software (version 5.0) (Thermo Fisher Scientific, Life Technologies Italia, Monza, Italy) was used for instrument control as well as data collection and processing. Data were acquired at the following wavelengths: 272 nm for theobromine and caffeine; 278 nm for (−)-epicatechin, catechin, type-B and type-A procyanidins; 293 nm for protocatechuic acid; and 350 nm for flavonol-3-O-glucosides (quercetin and kaempferol). The quantification was assessed by the external standard method using 6-point regression curves constructed at molecule maximum absorbance wavelength (R^2^ = 0.999).

### 2.5. Total Phenolic, Tannin, and Flavonoid Contents

The amount of total phenolics (TPC), flavonoids (TFC), and tannins (TTC) in CBS and CBS-IC was quantified as previously reported [33].

The BioTek Synergy HT spectrophotometric multi-detection 96-well microplate reader (BioTek Instruments, Milan, Italy) was used to record the absorbance of each fraction. All the analyses were performed in triplicate from 3 independent experiments. The concentration of total phenolic compounds, expressed in mg of gallic acid equivalents (GAE)/L, was determined using a standard curve of gallic acid (20–100 mg/L). TFC and TTC were expressed in mg of catechin equivalents (CE)/L referring to a standard curve of catechin (5–500 mg/L).

### 2.6. Antioxidant Capacity

CBS fraction radical scavenging activity (RSA) was evaluated by DPPH radical-scavenging assay. RSA was estimated by recording DPPH absorbance at 517 nm (using a 96-well microplate reader as reported above) and the analyses were performed in triplicate from 3 independent experiments. The inhibition percentage (IP) of DPPH radical was calculated using the following equation: IP (%) = [(A0−A30)/A0] × 100 (where A0 is the absorbance at time 0, and A30 the absorbance after 30 min). A standard curve of Trolox was used (12.5–300 μM) to assess the radical-scavenging activity values, which were expressed as µmoles of Trolox equivalents (TE) for mL of sample (µmol TE/mL).

### 2.7. Cell Culture and Treatments

The Cell Bank Interlab Cell Line Collection (Genoa, Italy) provided human colorectal adenocarcinoma CaCo-2 cells (accession number: ICLC HTL97023). As indicated on the Cell Bank Website (http://www.iclc.it/details/det_list.php; [34]), cells were plated at 1 × 10^6^/mL density, cultured in DMEM supplemented with 10% heat inactivated FBS, 1% antibiotic/antimycotic solution (100 U/mL penicillin, 0.1 mg/mL streptomycin, 250 ng/mL amphotericin B and 0.04 mg/mL gentamicin), and maintained at 37 °C in a humidified atmosphere containing 5% CO2. To allow their spontaneous differentiation into enterocyte-like phenotype, cells were grown for additional 18 days after reaching confluence [35].

CaCo-2 differentiation grade was routinely performed by monitoring the alkaline phosphatase (ALP) enterocyte brush border enzyme [36]. The cell monolayer was lysed by incubating cells at room temperature (RT) for 1 h with 300 µL PBS with 0.5% Triton X-100 (*v/v*). The lysates were dispensed into 96-well plate (corresponding to 5 × 10^5^ cells), and 60 µL of enzyme substrate (0.83 M 2-amino-2-methyl-1-propanol, 3.3 mM MgCl2, 13.3 mM disodium p-nitrophenol phosphate pH 10) was added to each well. After 30 min at 37 °C, the formation of p-nitrophenol phosphate cleaved product was stopped by adding 100 µL 0.5 M NaOH to the wells. The product absorbance was recorded with a 96-multiwell plate reader (Model 680 Micro-plate Reader, Bio-Rad laboratories Srl, Milan, Italy) using a 405 nm wavelength filter. ALP activity was expressed as nmoles of p-nitrophenol phosphate produced/mg cell protein. Proteins were evaluated with Bio-Rad protein assay dye reagent, following the protocol published by Bradford [37]. Table 1 shows the time course of ALP activity (microscopy images of CaCo-2 cells at confluence and differentiated CaCo-2 cells after 18 days post-confluence are shown in Appendix A).

After cell differentiation, CaCo-2 cells were incubated in serum-free medium overnight to make them quiescent before treatment. Cells were then treated at 37 °C for 24 h (or 6 h for gene expression experiments) with an oxysterol mixture (Oxy-mix) at 60 µM final concentration, and 5% FBS was added to DMEM to allow oxysterol uptake by cells.

The percentage composition of oxysterols used in the Oxy-mix was 42.96% for 7 K, 32.3% for α-epox, 5.76% for β-epox, 4.26% for 7α-HC, and 14.71% for 7β-HC. The molarity of each oxysterol in 60 µM Oxy-mix was calculated as 25.8 µM 7K, 19.4 µM α-epox, 3.4 µM β-epox, 2.6 µM 7α-HC, and 8.8 µM 7β-HC by considering an average molecular weight of 403 g/mol [30].

The 18-day differentiated CaCo-2 cells were preincubated with CBS or CBS-IC extracts for 1 h before the Oxy-mix treatment. Based on cell death analyses, CBS or CBS-IC were used to reach final 5% concentration in the cell culture.

In some experiments, cells were pretreated with 13 µM theobromine corresponding to its concentration in CBS-IC, and similar analyses as for CBS/CBS-IC in the presence or not of Oxy-mix were done.

### 2.8. Cell Death Evaluation

The extracellular release of lactate dehydrogenase (LDH) was considered as a parameter of cell death. Cells were treated with IC, CBS-IC, or CBS, increasing concentrations to reach their final percentage in culture media of 5%, 10%, 30%, and 50% (*v/v*), and incubated or not with 60 µM Oxy-mix. Untreated cells incubated for the same time period as treated cells were considered as control.

LDH was evaluated spectrophotometrically at 340 nm wavelength by recording NADH production/min. LDH release of 3 independent experiments was performed in triplicate and expressed as a percentage of the total enzyme released into cell culture medium by complete cell lysis (obtained by 0.5% Triton X-100 addition to the plate containing the same cell density as the treated cells) (Appendix A).

### 2.9. Evaluation of Interleukin-8 and Monocyte Chemoattractant Protein-1 Protein Levels by Enzyme-Linked Immunosorbent Assay

IL-8 and MCP-1 levels released by the cells into the medium were evaluated in the culture media from treated differentiated CaCo-2 cell samples. Controls were referred to untreated differentiated cells incubated with the same cell culture medium quantity used for treatments (5% FBS-DMEM) at the same times as treated differentiated cells. Cytokines were analyzed and estimated using commercial ELISA kits. Capture antibody anti-human MCP-1 or anti-human IL-8 was added to each ELISA plate well (0.25 µg/mL concentration for MCP-1 and 0.125 µg/mL concentration for IL-8, following manufacturer’s instruction) and incubated overnight. Afterward, the plate wells were washed thrice with the Wash Buffer (0.05% Tween-20 in phosphate buffered saline, PBS), incubated 1 h at RT with the Block Buffer (1% bovine serum albumin (BSA) in 1X PBS), and washed thrice with the Wash Buffer. The samples were then added to the plate wells, incubated for 2 h at RT, and finally washed 4 times with the Wash Buffer. The detection antibody was added to the plate wells at a 0.25 µg/mL concentration and incubated for 2 h at RT, then washed 4 times with the Wash Buffer. Streptavidin-HRP conjugate was added to the plate wells at a 0.05 µg/mL concentration for 30 minutes at RT. After 4 washes with the Wash Buffer, the plate wells were incubated with 3,3′,5,5′-tetramethylbenzidine (TMB) Liquid Substrate for 20 minutes at RT, and the color development was blocked with the stop solution (1 M HCl). Color development was monitored with a 96-multiwell plate reader (Model 680 Microplate Reader, Bio-Rad laboratories Srl, Milan, Italy) using a 450 nm wavelength filter. Absorbance at 655 nm wavelength was considered as a value reference for each sample.

The Bio-Rad protein assay dye reagent was used to estimate the total protein concentration in each sample cell medium [37]. IL-8 and MCP-1 levels were normalized to proteins of the incubation media and values expressed as pg cytokines/mg cell culture medium proteins.

The analyses were performed in triplicate from 3 independent experiments, and data were calculated using SlideWrite Plus software (Advanced Graphics Software, Rancho Santa Fe, CA, USA).

### 2.10. Tight-Junction Protein Immunoblotting

At the end of each treatment, differentiated cells were scraped and washed with ice-cold PBS 1X, prepared by diluting 10X PBS (containing 1.37 M NaCl, 27 mM KCl, 100 mM Na2HPO4, and 18 mM KH2PO4) in Milli-Q water. Then, 150 μL lysis buffer were added for protein extraction (147 μL PBS supplemented with 1.5 μL Triton X-100 (*v/v*), 1.5 μg sodium dodecyl sulfate (SDS) (*w/v*) (final volume)). Once lysed, samples were incubated for 30 min on ice and centrifuged at 12,052× *g* at 4 °C for 15 min. Total cell extract protein concentration was evaluated with Bio-Rad protein assay dye reagent following the protocol published by Bradford [37].

Each sample containing 50 µg total proteins was boiled at 100 °C in the Sample Buffer (LDS Sample Buffer 4X and DTT Sample Reducer 10X) for 5 min. Boiled samples were subjected to electrophoresis separation using 10% SDS-polyacrylamide gel, and proteins were transferred to Hybond ECL nitrocellulose membranes. The membranes were then incubated at RT for 1 h in 10 mL TBS supplemented with 5 µL Tween 20 (TTBS) blocking buffer plus 0.5 g skimmed milk powder. Blots were then incubated at 4 °C overnight with either 10 mL mouse anti-claudin 1 (1:800 dilution), 10 mL mouse anti-occludin (1:500 dilution), or 10 mL rabbit anti-JAM-A (1:200 dilution) polyclonal antibodies in TBS containing 10 µL Tween-20 and 0.5 g skimmed milk powder. Three subsequent washes in TTBS were performed, and blots were incubated with anti-rabbit or anti-mouse HRP-conjugated IgG (1:1000 dilutions) in 10 mL TBS with 10 µL Tween-20 and 0.5 g skimmed milk powder for 1 h. Finally, blots were washed twice in TTBS for 10 min.

The Clarity Western ECL kit and ChemiDoc™ Touch Imaging System machine (Bio-Rad laboratories Srl, Segrate, Italy) were used to detect chemiluminescence. Protein band densities of the 3 performed independent experiments were quantified using Image J Software (Bethesda, MD, USA).

### 2.11. Tight Junction Protein Immunofluorescence

Cell localization of TJ proteins (claudin 1, occludin, and JAM-A) was visualized using immunofluorescence technique. For this analysis, treatments were performed using differentiated CaCo-2 cell monolayers grown on 13 mm diameter glass slides. After treatments, slides were washed twice with PBS, fixed with 4% paraformaldehyde for 15 min, and then permeabilized with 0.2% Triton X-100 in PBS at RT for 5 min. Two PBS washes and incubations at RT for 30 min with a blocking buffer (PBS containing 5% goat serum, 3% BSA, and 0.3% Tween) were performed to prevent nonspecific binding of the primary antibodies. After 2 further washes with PBS, CaCo-2 cell slides were incubated overnight at 4 °C with mouse anti-occludin (1:50 dilution), mouse anti-claudin 1 (1:50 dilution), or rabbit anti-JAM-A (1:100 dilution) antibodies. Regarding claudin 1 and occludin visualization, cells were then washed and incubated with goat anti-mouse IgG-Alexa Fluor 488 (1:500 dilutions) at RT for 2 h (antibody dilution was performed in PBS containing 0.5% BSA and 0.1% Triton X-100). Regarding JAM-A visualization goat anti-rabbit IgG-Alexa Fluor 546 was used for incubation as described above.

After 2 further washes, slides were mounted with Fluoroshield mounting medium and observed using a LSM 800 confocal laser microscope (Zeiss SpA, Oberkochen, Germany).

Excitation values of 488 nm and emission values of 505–550 nm were used for green fluorescence, while excitation values of 546 nm and emission values of 573 nm were used for red fluorescence. Images were processed using LSM 800 Image Examiner software (Zeiss SpA, Oberkochen, Germany).

### 2.12. Real-Time Quantitative Reverse-Transcription Polymerase Chain Reaction 

Differentiated cells were pretreated or not with CBS, CBS-IC for 1 h and incubated with Oxy-mix for 6 h. This incubation time corresponds to the significant maximum Nrf2 expression. In a group of experiments, cells were pretreated with 13 µM theobromine corresponding to its content in CBS-IC; pretreatments with 1 µM (−)-epicatechin were also inserted as positive controls.

Total mRNA was extracted from treated cells using 1 mL TRIzol™ following manufacture instructions and protocol published by Rio and colleagues [38]. mRNA extract concentrations were detected by measuring ribonucleic acid absorbance at 260 nm. Ultraviolet (UV) absorbance ratio at 260/280 nm was used to assess RNA purity. Reverse transcription was performed to synthesize cDNA from 2 μg mRNA using random primers and following the manufacturer’s instructions of a commercial kit. Real-time quantitative reverse-transcription polymerase chain reaction (qRT-PCR) was performed on 20 ng cDNA using TaqMan gene expression probes for Nrf2, and amplified cDNAs were analyzed using the 7500 Fast real-time PCR system (Applied Biosystems, Thermofisher, Monza, Italy). Oligonucleotide sequences have not been disclosed by the manufacturer because they are strictly confidential. PCR’s cycling parameters were 40 cycles/3 s each at 95 °C (melting) and 30 s at 60 °C (annealing/extension). Results were normalized to the expression of β-actin as a housekeeping gene. Target gene expression was quantified as proposed by Livak and Schmittgen [39].

### 2.13. Statistical Analyses

Statistical differences among data regarding molecular experiments, performed as 3 independent experiments, were evaluated using the 1-way ANOVA test associated with Bonferroni’s multiple comparison post-test.

Differences among values obtained from chromatographic analyses were evaluated by ANOVA associated with Duncan’s test, being, in this case, the most suitable test in multiple comparison. 

All values were expressed as mean ± Standard Deviation (SD) of 3 independent experiments, and data were analyzed with GraphPad InStat software (San Diego, CA, USA).

## 3. Results

### 3.1. Cocoa Bean Shell-Enriched Ice Cream Shows High Recovery Content of Phenolic and Methylxanthines and Antioxidant Capacity

CBS flour addition to plain ice cream gave important quantitative changes in both phenolic compounds and xanthine derivatives (Table 2 and Table 3). Catechin with its glucoside isomers, (−)-epicatechin and procyanidins, as well as theobromine and caffeine, displayed the highest concentration values in CBS-IC samples. Methylxanthines were the main compounds in all samples. Theobromine was one of the main compounds identified in high concentrations (47.47 µg/mL) in the fraction obtained after complete gastrointestinal digestion of CBS-IC (Table 2).

Notably, RP-HPLC-PDA analyses performed in the different formulations for the detection of total phenolics, tannins, and flavonoids showed the highest recovery in the CBS-IC sample. Increased antioxidant capacity was also found in CBS-IC (Table 3).

### 3.2. Cocoa Bean Shell and Cocoa Bean Shell-Enriched Ice Cream Prevent Interleukin-8 and Monocyte Chemoattractant Protein-1 Production from Differentiated CaCo-2 Cells Treated with the Proinflammatory Oxysterol Mixture

Differentiated CaCo-2 cells were pretreated (1 h) with CBS-IC or CBS alone and then incubated with 60 µM dietary Oxy-mix for 24 h. Based on the cytotoxicity analyzed as LDH release, 5% CBS-IC/CBS (0.150 g in 3 mL cell culture final concentration) showed no cytotoxic effects both in presence and absence of the Oxy-mix and was therefore chosen as the best concentration for cell treatment.

CBS-IC and CBS anti-inflammatory efficacy was evaluated in terms of IL-8 and MCP-1 production.

As expected, Oxy-mix induced a strong increase in the proinflammatory cytokines released by CaCo-2 cells in the culture medium (2.25- and 3.3-fold increase of IL-8 and MCP-1, respectively) (Figure 1).

Cell pretreatment with the different CBS samples decreased IL-8 and MCP-1 overproduction induced by the Oxy-mix. Although both cytokine levels clearly decreased in CaCo-2 cells pretreated with CBS/CBS-IC compared with the only Oxy-mix treatment, the highest significance was evident in MCP-1 production (Figure 1B). Data from cell pretreatments in absence of the Oxy-mix showed that neither CBS nor CBS-IC influenced IL-8 and MCP-1 medium basal levels.

According to these findings suggesting CBS can add value to plain ice cream as a source of bioactive compounds with antioxidant properties, CBS and CBS-IC extracts were chosen to perform the in vitro experiments in CaCo-2 cell monolayers.

### 3.3. Tight Junction Impairment Induced by the Oxysterol Mixture Is Prevented in Differentiated CaCo-2 Cells Pretreated with Cocoa Bean Shell-Enriched Ice Cream and Cocoa Bean Shell 

To elucidate the potential beneficial effect of CBS formulations in maintaining the intestinal epithelial barrier integrity, we examined protein levels and cellular distribution of the main TJs in differentiated CaCo-2 cell monolayers. Claudin 1, occludin, and JAM-A were studied as TJs involved in the regulation of the layer permeability.

Differentiated CaCo-2 cell monolayers were incubated with 60 µM Oxy-mix for 24 h. Western Blotting analyses showed that Oxy-mix induced a significant decrease of all TJ levels, such a deranging effect being remarkable on JAM-A (Figure 2C). Immunofluorescence analysis underlined a clear TJ cellular delocalization after the Oxy-mix cell treatment (Figure 3). On the other hand, cell pretreatment with CBS-IC or CBS fully prevented the Oxy-mix-exerted decrease of TJ protein levels (Figure 2).

Interesting results came from the analyses on TJ cellular distribution carried out by fluorescence microscopy. CBS-IC and CBS pretreatment contributed to maintain a correct cell membrane assembly of claudin and occludin, although JAM-A spatial Oxy-mix-dependent delocalization appeared not to be completely prevented by CBS-IC (Figure 3).

### 3.4. Cocoa Bean Shell Can Affect Antioxidant Capacity of Differentiated CaCo-2 Cells Treated with Oxysterol Mixture by Increasing Nrf2 Gene Expression

The antioxidant capacity observed for the CBS formulations led us to explore the in vitro cell potential effect of these extracts in modulating the cell adaptive response other than their direct scavenging activity. 

An upregulation of Nrf2 mRNA expression was already evident when differentiated CaCo-2 cells were treated with the only Oxy-mix, demonstrating the ability of dietary oxysterols to induce the cellular antioxidant defense system if added in mixture to the cells (Figure 4). Cells treated with CBS or CBS-IC alone also showed an increase in Nrf2 gene expression, particularly CBS-IC. A significant fold increase was evident in cells treated with CBS-IC + Oxy-mix compared with cells incubated with the Oxy-mix alone (Figure 4).

### 3.5. Theobromine Can Preserve Oxysterol-Mediated Tight Junction Derangement and Inflammation but Does Not Reinforce Antioxidant Cell Response

We aimed at elucidating the possible contribution of theobromine in protecting cell layer damage induced by 60 µM Oxy-mix. This methylxanthine represents the major CBS component besides flavonoids. Therefore, differentiated CaCo-2 cell monolayers were pretreated with 13 µM theobromine for 1 h and then treated with the Oxy-mix. This concentration corresponds to the highest theobromine content evaluated in CBS-IC (Table 2). IL-8 and MCP-1 levels released by the cells in the culture medium, TJ cellular distribution, and Nrf2 gene expression were evaluated.

Theobromine did not influence IL-8 and MCP-1 basal levels but was able to prevent chemokine increase observed in Oxy-mix treated cells (Figure 5A,B).

Immunofluorescence analyses on claudin 1, occludin, and JAM-A cellular distribution showed that correct cell membrane TJ localization was maintained by theobromine cell pretreatment (Figure 5C). On the other hand, Nrf2 gene expression in the Oxy-mix treated cells was not upregulated by theobromine. Only 1 µM (−)-epicatechin cell treatments considered as positive controls showed a significant increase compared with Oxy-mix treated cells (Figure 5D).

## 4. Discussion

In the last decade, an altered intestinal permeability has been associated with the pathogenesis of human disorders such as inflammatory bowel diseases [20]. Intestinal mucosal damage results from tight junction loss, increased paracellular transport, apoptosis, and transcellular permeability [40,41].

A leaky gut may be a crucial event allowing digestive metabolites and bacteria-derived molecules to enter the mucosa and trigger inflammatory processes, thus inducing/amplifying the barrier damage [42].

Western dietary habit changes, particularly rich in fats of animal origin such as cholesterol and its oxidation products, namely oxysterols, represent one of the main risks for altered intestinal integrity [24,26].

Various studies on oxysterol percentage distribution detected in foods in relation to dietary cholesterol intake are now available [43,44,45].

In the present study, the CaCo-2 cell monolayer was treated with the same mixture and concentration (60 µM) of the main oxysterols deriving from cholesterol auto-oxidation in cholesterol-rich foods, which were employed in the previous studies as a reliable experimental model of barrier damage by exerting remarkable pro-oxidant and proinflammatory action [27].

The alteration of cellular permeability and tight-junction protein zonula occludens-1 was observed in CaCo-2 cells cocultured with dendritic cells in the presence of 7K, as well as in vascular smooth muscle cells and monocytes treated with 7K, 7α-HC, or 7β-HC [46,47]. These cholesterol-oxygenated derivatives, which are the main constituents of cholesterol-rich foods, are also present in the atherosclerotic plaques [46,47].

As we demonstrated previously, the dietary oxysterol mixture (where 7K, 7α-HC and 7β-HC are mainly present) affects the permeability of the CaCo-2 cells grown as a monolayer. In these cells, the mixture was found to increase the activity of matrix metalloproteinases 2 and 9, which were responsible for intestinal barrier destabilization [22]. This increase is probably due to the oxysterol capacity to trigger inflammation through the activation of Toll like receptors TLR2 and TLR4. The main ligands of these receptors are bacterial lipopeptide or lipopolysaccharide, whose chemical structure is likely to be of lipid- or oxidatively-modified lipid-origin similarly to oxysterols [30]. The action of dietary oxysterols to directly alter TJ cellular levels and distribution was also underlined. These compounds may displace cholesterol in lipid raft microdomains where TJs are organized, thus influencing TJ compartmentalization and function [22,48].

The cocoa bean shell, arising from the remnants of the chocolate production process, has recently attracted attention as a potential raw material to be employed in food supplements [8].

Our findings demonstrate the biological impact of CBS, confirming its antioxidant and anti-inflammatory action in our experimental model of enterocyte-like cell culture. Cell pretreatment with CBS or CBS-enriched ice cream significantly prevented 60 µM Oxy-mix-dependent inflammation in terms of decreased IL-8 and MCP-1 cell release in the culture medium, two chemokines whose increase contributes to the intestinal mucosal damage under proinflammatory conditions, e.g., bacteria toxin exposure [49]. Consistently, both CBS extracts restored protein levels of the main TJs involved in the regulation of paracellular permeability claudin 1, occludin, and JAM-A, which were altered in oxysterol-treated CaCo-2 cells.

Claudins, occludin, and JAM-A cooperate in physical and functional paracellular permeability maintenance. Besides this function, JAM-A is involved in the leukocyte recruitment during inflammation [50]. Along TJ complex remodeling, occludin and claudins merely spread from the membrane to the cytosolic fraction, whereas JAM-A is transiently stored in endosomes and redistributed toward the cell sites, which are likely to be more accessible to mediate leukocyte interaction. Stamatovic and colleagues thoroughly described JAM-A re-localization in the brain endothelium as a mechanism enabling leukocyte adhesion under inflammatory conditions [51].

Regarding the effect of CBS and CBS-IC in cellular TJ distribution in our experimental model, differentiated CaCo-2 cells treated with Oxy-mix but pretreated with either CBS or CBS-IC showed similar claudin 1 and occludin cellular distribution to the control. However, JAM-A was still partially spread in CBS-IC pretreated cells. Such incomplete recovery could be due to ice cream fat content in CBS-IC sample impeding proper JAM-A endosome redistribution despite its protein content recovery.

The beneficial property of CBS in preserving intestinal cell layer from permeability alteration, suggested by the above data, could be due to the high concentration of antioxidant polyphenols which are present in this byproduct [33] and that have been recently emerged for their potential modulatory properties with respect to intestinal permeability [18]. (-)-Epicatechin was shown to prevent intestinal barrier permeabilization and inflammation both in vitro in TNFα-treated CaCo-2 cell monolayers and in vivo in male C57BL/6J mice fed with a high-fat diet through a negative modulation of NADPH oxidase, reactive oxygen species production, and NF-κB activation [52,53]. This signaling pathway is the main one induced by dietary oxysterols in CaCo-2 monolayers [27] that are able to trigger inflammation and derange intestinal permeability [22]. The TJ level decrease and inflammatory cytokine release observed in the cells treated with the same mixture of oxysterols as used in the present study was found to be preserved by 1 µM (−)-epicatechin [22].

Polyphenols, including (−)-epicatechin, have been shown to be potent activators of the cellular resistance regulator to oxidants Nrf2 [54,55]. In our present study, we demonstrated not only the ex vitro CBS antioxidant capacity, but also its potential biological impact to upregulate Nrf2 gene expression when cells were exposed to dietary oxysterols. We used (−)-epicatechin in the same concentration, which was able to prevent inflammation and TJ derangement as previously described [22], and considered it as a positive control. The oxysterol mixture was able to slightly increase Nrf2 expression, probably in response to its property to induce an intracellular oxidant environment. Interestingly, CBS-enriched ice cream, which was characterized by higher flavonoid content than CBS, showed the highest capability to induce Nrf2 gene expression, and significantly increased its oxysterol-dependent activation. This finding highlights CBS health benefits as polyphenol-enriched functional food ingredient.

However, we cannot exclude the beneficial role of other compounds than polyphenols, which are highly present in CBS. A relevant content of methylxanthines in cocoa, especially in dark chocolate from Forastero cultivar, has been known for a long time [56]. Methylxanthines, especially theobromine, are particularly abundant in CBS because during cocoa bean fermentation they migrate into the shell [12]. Theobromine has recently been considered as a much more efficient molecule than caffeine in exerting anti-inflammatory/antitumor effects, and cardiovascular protection [16]. In fact, theobromine seems to show differential and weaker action compared with its chemical homologous caffeine, avoiding the unpleasant caffeine-associated effects like anxiety or insomnia. The lower affinity for adenosine receptors showed by theobromine compared with caffeine may be responsible for the observed different psychoactive effects of theobromine [57]. CBS compound characterization showed a high theobromine content in CBS and reaches the highest concentration in CBS-IC.

Results on theobromine effects concerning its role in the modulation of chemokines and TJ protein cell production and distribution show that this molecule, added to the cells in the same concentration as CBS-IC formulation, prevented Oxy-mix-dependent inflammation and TJ derangement. Theobromine activity in inhibiting phosphodiesterase was suggested as a mechanism of action to reduce obesity and related adipose tissue inflammation in mice [58]. Anti-inflammatory and antioxidant effects of theobromine through NF-kB downregulation were observed in chondrocytes treated with IL-1β [59]. The anti-inflammatory effect of theobromine, even in lower concentration than that used in this paper, agrees with recently published data [19]. However, theobromine was unable to up-regulate Nrf2 gene expression, suggesting that this molecule does not show antioxidant capacity at least in our biological system.

## 5. Conclusions

The reported data underline the ability of CBS and CBS-enriched ice cream in preserving the intestinal cell monolayer from the deranging action of an oxysterol mixture detectable in cholesterol rich foods in terms of reduced inflammation, TJ protein loss, and redistribution, as well as stress-related antioxidant response potentiation. Various studies have suggested cocoa bean shell reuse in functional food because of its richness in polyphenols’ potential benefit on health [8,33].

The analysis of CBS-enriched ice cream extract showing higher flavonoid content than plain ice cream strengthens the actual possibility to apply this byproduct as a food ingredient. However, we would like to point out that theobromine accumulates in CBS in very high concentration. Although further studies are required to clarify the mechanisms by which theobromine can protect the intestinal layer from dietary attacks, a cooperating—even though partial—role in CBS biological activity cannot be excluded.

## Figures and Tables

**Figure 1 antioxidants-10-00280-f001:**
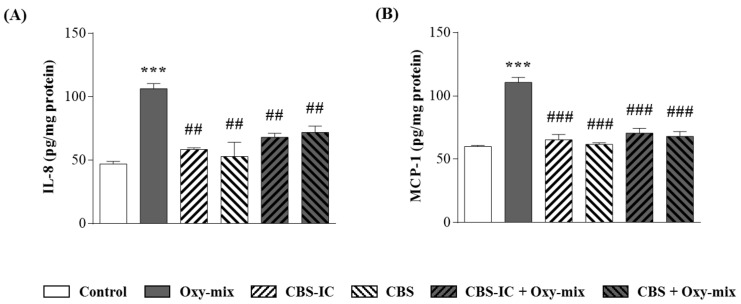
Oxy-mix-mediated inflammation was prevented by CBS extracts. IL-8 and MCP-1 levels ((**A**) and (**B**), respectively) were measured by ELISA in the cell medium of differentiated CaCo-2 cells pretreated or not with 4% CBS-enriched ice cream (CBS-IC) or CBS alone for 1 h and incubated with 60 µM oxysterol mixture (Oxy-mix) for 24 h. Both pretreatments were able to fully prevent the Oxy-mix-dependent cytokine protein level increase in the cell medium. Values are shown as pg chemokines normalized for cell incubation medium mg proteins (pg/mg protein). Values are referred as means of three independent experiments (*n* = 3) evaluated in triplicate ± SD. Significantly different vs. controls: *** *p* < 0.001; significantly different vs. Oxy-mix: ## *p* < 0.01, ### *p* < 0.001.

**Figure 2 antioxidants-10-00280-f002:**
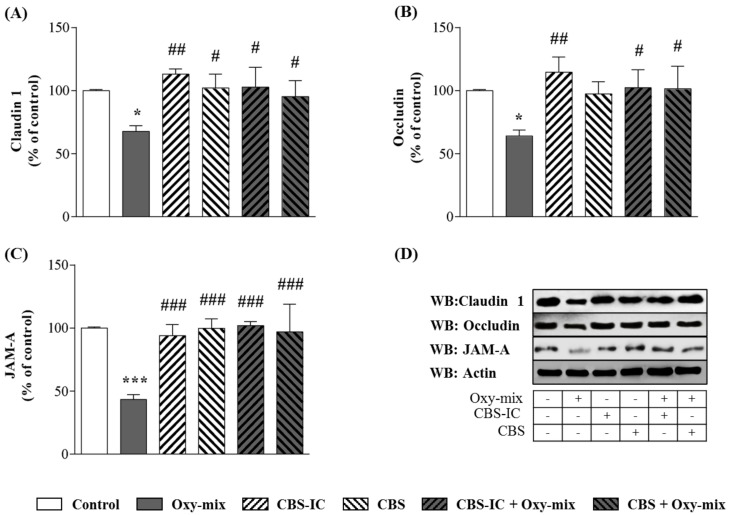
CBS and CBS-enriched ice cream prevented TJ loss induced by Oxy-mix. Decreased levels of claudin 1 (**A**), occludin (**B**) and JAM-A (**C**) were detected by Western Blotting in lysates from CaCo-2 cells incubated with 60 µM Oxy-mix for 24 h. One-hour CBS or 4% CBS-enriched ice cream (CBS-IC) pretreatment protected oxysterol-mediated TJ loss. A representative Western Blot of each treatment is shown (**D**). Data are expressed as percentage of control (100%). Values are means ± SD of three independent experiments. Significantly different vs. controls: * *p* < 0.05, *** *p* < 0.001; significantly different vs. Oxy-mix: # *p* < 0.05, ## *p* < 0.01, ### *p* < 0.001.

**Figure 3 antioxidants-10-00280-f003:**
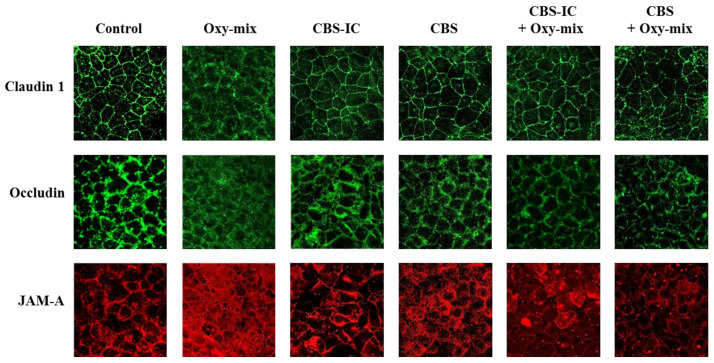
CBS formulations prevented TJ delocalization observed in oxysterol-treated CaCo-2 cell monolayers. A marked redistribution of claudin 1, occludin, and JAM-A after cell monolayer incubation with 60 µM Oxy-mix for 24 h was visualized by immunofluorescence microscopy. CBS-IC or CBS pretreatments restored cellular localization, particularly claudin 1 and occludin. Claudin 1 and occludin images were immunostained with green color, whereas red color was used for JAM-A immunostaining (see Materials and Methods). Original magnification: ×400.

**Figure 4 antioxidants-10-00280-f004:**
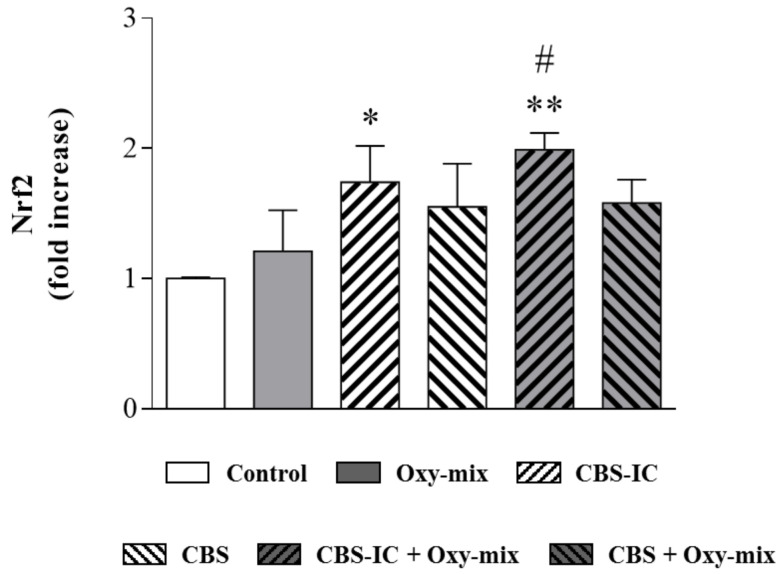
Oxysterol-dependent Nrf2 gene expression was upregulated by CBS-enriched ice cream. Nrf2 gene expression was evaluated by qRT-PCR in differentiated CaCo-2 cells treated with 60 µM Oxy-mix for 6 h, and pretreated or not with CBS-IC or CBS alone for 1 h. Data are expressed as fold induction than control sample. Preincubation with CBS-IC showed a significant fold increase compared with Oxy-mix treatment. Data are reported as means of three independent experiments (*n* = 3) evaluated in triplicate ± SD. One-way analysis of variance (ANOVA) associated with Bonferroni’s multiple comparison post-test was adopted: Significantly different vs. controls: * *p* < 0.05, ** *p* < 0.01; significantly different vs. Oxy-mix: # *p* < 0.05.

**Figure 5 antioxidants-10-00280-f005:**
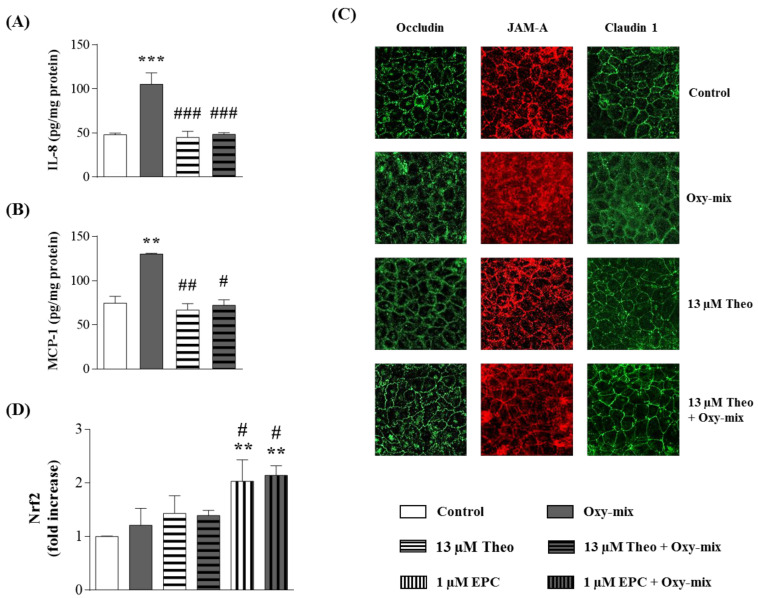
Theobromine prevented IL-8, MCP-1 and TJ changes but did not increase Nrf2 gene expression in differentiated CaCo-2 cells treated with Oxy-mix. Differentiated CaCo-2 cells were pretreated or not with 13 µM theobromine for 1 h and incubated with 60 µM Oxy-mix for 24 h. Cell release of IL-8 (**A**) and MCP-1 (**B**) in the incubation medium was detected by ELISA and evaluated. Values are shown as pg chemokines normalized for cell incubation medium mg proteins (pg/mg protein). Data are means of three independent experiments (*n* = 3) evaluated in triplicate ± SD. Claudin 1, occludin, and JAM-A cellular distribution (**C**) was evaluated in differentiated CaCo-2 cell lysates in presence of 13 µM theobromine (1 h cell pretreatment) incubated or not with 60 µM Oxy-mix for 24 h. Fluorescence microscopy was used to visualize TJ cellular distribution (×400 magnification). Claudin 1 and occludin proteins were visualized as green fluorescence, JAM-A was visualized as red fluorescence (see Materials and Methods). Nrf2 gene expression (**D**) was evaluated through qRT-PCR in differentiated CaCo-2 cells after 6 h Oxy-mix incubation in presence or absence of 13 µM theobromine. The histograms represent means of three independent experiments (*n* = 3) evaluated in triplicate ± SD, normalized to the corresponding housekeeping gene actin and expressed as fold induction vs. control. Theo: theobromine; EPC: (−)-epicatechin. Significance: Different vs. controls: ** *p* < 0.01, *** *p* < 0.001; different vs. Oxy-mix: # *p* < 0.05, ## *p* < 0.01, ### *p* < 0.001.

**Table 1 antioxidants-10-00280-t001:** Time course of alkaline phosphatase activity in CaCo-2 cells after reaching cell confluence.

Post-Confluence Days	Alkaline Phosphatase Activity (nmoles/mg Protein)
**T0**	39.18 ± 10.2
**T3**	83.84 ± 6.7 *
**T6**	287.29 ± 22.8 ***
**T9**	329.87 ± 17.6 ***
**T15**	437.80 ± 15.9 ***
**T18**	450.55 ± 18.4 ***

Values of alkaline phosphatase activity are shown as nmoles/mg protein and are referred as means ± SD of 6 independent experiments. Cell cultures were grown up to 18 days after confluence (T0). T3, T6, T9, T15, and T18 refer to post-confluence days of cell culture (3, 6, 9, 15, 18 days, respectively). Significantly different vs. T0: * *p* < 0.05, *** *p* < 0.001.

**Table 2 antioxidants-10-00280-t002:** Main components identified and quantified by high-performance liquid chromatography (HPLC) in solutions yielded after gastrointestinal digestion of plain ice cream (IC), ice cream enriched with 4% CBS (CBS-IC), and CBS (CBS).

Compound (µg/mL Extract)	IC			CBS-IC			CBS			Sig
Theobromine	18.69	±	0.65 ^c^	47.47	±	0.59 ^a^	36.6	±	2.78 ^b^	***
Caffeine	1.11	±	0.26 ^c^	6.49	±	0.09 ^a^	5.99	±	0.09 ^b^	***
Protocatechuic acid	0.06	±	0.01 ^a^	0.22	±	0.01 ^a^	0.16	±	0.16 ^a^	n.s.
N-Coumaroyl-L-aspartate_isomer 1	0.11	±	0.01 ^c^	0.99	±	0.04 ^a^	0.65	±	0.12 ^b^	***
Catechin-3-O-glucoside_isomer 1	0.64	±	0.02 ^c^	1.83	±	0.04 ^a^	0.44	±	0.01 ^b^	***
Catechin	0.19	±	0.01 ^c^	1.05	±	0.03 ^a^	0.58	±	0.03 ^b^	***
Catechin-3-O-glucoside_isomer 2	0.35	±	0.08 ^c^	1.07	±	0.05 ^a^	0.71	±	0.02 ^b^	***
Epicatechin	0.00	±	0.00 ^b^	0.89	±	0.10 ^a^	0.87	±	0.10 ^a^	***
Procyanidin (PC)C	0.60	±	0.06 ^b^	1.28	±	0.05 ^a^	0.47	±	0.02 ^c^	***
PCA pentoside_isomer 1	0.33	±	0.05 ^b^	0.50	±	0.06 ^a^	0.00	±	0.00 ^c^	***
PCA pentoside_isomer 2	0.16	±	0.02 ^c^	0.84	±	0.03 ^a^	0.53	±	0.02 ^b^	***
Kaempferol-3-rutinoside	0.00	±	0.00 ^c^	0.17	±	0.01 ^a^	0.12	±	0.02 ^b^	***
Quercetin-3-arabinoside	0.00	±	0.00 ^c^	0.19	±	0.01 ^a^	0.03	±	0.02 ^b^	***

Data are expressed as mean values of three independent experiments (*n* = 3) evaluated in triplicate ± SD. Sig: Significance. ANOVA associated Duncan’s test was used as a multiple range test to highlight the significant differences among all the samples (IC, CBS-IC, and CBS). ANOVA significance: *** *p* < 0.001. Means showing different lowercase letters within the same line are significantly different at *p* < 0.05.

**Table 3 antioxidants-10-00280-t003:** Total phenolic, flavonoid, and tannin content and antioxidant capacity in IC, CBS-IC, and CBS.

	IC			CBS-IC			CBS			Sig
TPC (µg GAE/mL)	349.01	±	8.32 ^b^	476.69	±	7.09 ^a^	188.85	±	5.80 ^c^	***
TFC (µg CE/mL)	47.69	±	3.62 ^b^	96.06	±	7.73 ^a^	44.18	±	7.06 ^b^	***
TTC (µg CE/mL)	52.77	±	4.15 ^b^	59.48	±	2.47 ^a^	34.24	±	1.86 ^c^	***
RSA (µmol TE/mL)	0.74	±	0.08 ^b^	1.12	±	0.10 ^a^	0.70	±	0.03 ^b^	***

Total phenolic content (TPC), total flavonoid content (TFC), total tannin content (TTC), and antioxidant capacity (RSA) were evaluated in plain ice cream (IC), 4% CBS-enriched ice cream (CBS-IC) and CBS powder (CBS) obtained after gastrointestinal digestion. TPC are expressed as gallic acid equivalent (GAE); TFC and TTC as catechin equivalent (CE); RSA as Trolox equivalent (TE)/mL extracts. Analyses were performed in triplicate. Data are expressed as mean values of three independent experiments (*n* = 3) evaluated in triplicate ± SD. Sig: Significance. ANOVA associated Duncan’s test was used as a multiple range test to highlight the significant differences among all the sample (IC, CBS-IC, and CBS). ANOVA significance: *** *p* < 0.001. Means showing different lowercase letters within the same line are significantly different at *p* < 0.05.

## Data Availability

The data of this study are contained within the article or Appendix A.

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
