# Peer review of "Protective Effect of Cocoa Bean Shell against Intestinal Damage: An Example of Byproduct Valorization"

_antioxidants, 2021, doi:10.3390/antiox10020280_

Round 1

Reviewer 1 Report

The manuscript entitled „Protective Effect of Cocoa Bean Shell Against Intestinal Damage: an example of Byproduct Valorization” describes a very interesting study on the valorization of cocoa by-products as a potential food additive conferring health benefits.

The study idea is interesting, and a lot of work has been done. However, the manuscript itself needs a lot of improvements.

Title: Should be “by-products”

Abstract: First sentence is too long and gives the conclusion of this work, not the introduction. The abstract should contain the aim of the study. The ice cream enriched with cocoa bean shell were studied before, therefore they are in the introduction? Has the ice cream being designed in this study or taken from somewhere? What about theobromine? The whole abstract has to be rewritten.

Introduction:

Latin names should be in italic

Please explain all the abbreviations in the first use in the text. Names of chemicals and enzymes/proteins/markers too. Explanation in the abstract is not enough, both manuscript and abstract should be able to stand alone.

Line 45-53: Why each sentence is a separate paragraph?

Line 81-83: What recent study? Please provide a reference.

The sentence “The aim of our current investigation is to give in vitro further evidence of CBS positive effect against oxysterol-induced oxidative…” suggest that there are studies in this topic? Or do the authors refer to their research on red wine and olive oil?

The introduction is inconsistent and does not describe sufficiently neither research gap nor cocoa bean shell. Sentences are not connected. The introduction looks like a draft, not the final version. Please rewrite it.

Line 96: Have the authors used DPPH for HPLC? This is something new.

Line 135: Reference 23 does not have ice cream formulation. The reference is about pulsed electric field assisted extraction of bioactive compounds from cocoa bean shell.

I would suggest reorganizing Materials and Methods section. First describe the product formulation, its characteristic, then start with the in vitro experiment. Otherwise, it is messy and difficult to follow.

Results:

The authors wrote: “Chromatographic analyses of CBS and ice cream showed that CBS-IC maintains similar structural characteristics to the common chocolate ice cream, making it suitable for consumer acceptance [8]”. Firstly, how chromatographic analysis can inform about the structure? The texture analysis is required for this, which was not performed in this study. Secondly, again the authors cite themselves claiming findings which are not in their work. The reference 8 is about the beverage, not the ice cream, therefore it is not correct that the “chromatography-evaluated” structure of ice cream has something in common with reference 8.

The formulation of ice cream is an important factor, therefore it should be in the main text, not in the supplement.

Line 334: If the ice cream was supplemented with only 4% of CBC, then how “Theobromine reached about 50% of the whole CBS-IC composition”. 50% of the formulation was skimmed milk, not theobromine.

Does the plain ice cream have higher antioxidant capacity and bioactive compounds content that CBS? In this case, I think ice cream does not need fortification with less rich material.

Line 407: restored? I thought that CBS was applied first, before oxysterols. Restoring suggests that the CBS was applied after oxysterols. Which is the correct approach?

Discussion:

Line 524-526: If the effect on the intestinal permeability was related to flavonoids, why then the authors performed the additional experiment with theobromine, not with e.g. epicatechin?

Reviewer 2 Report

General comments:

  1. Several statements in the introduction lacked references.
  2. Material and methods were insufficiently detailed rendering difficult to assess the reliability and reproducibility of the findings by other laboratories.
  3. It was unclear why human colorectal adenocarcinoma CaCo-2 cell line was used. There was no characterization of the cells and it was unclear how cell differentiation, if any, was monitored.
  4. There was insufficient information on cell treatments, with respect to the state of differentiation of cells and duration of treatments.
  5. The overall findings appear preliminary due to the lack of assays on other different cell lines, or primary cells, inconclusive findings on the effects of theobromine as well as insufficient information on the bioactive compounds. Flavonoids were suspected to be beneficial but there were no experiments to assess their eventual beneficial effects.
  6. Several statements in the discussion could be merged into the introduction.
  7. There were too much self citations (10 over 46). References N° 8,16, 17,18,19,20,22,23,32 and 46.

Minor comments:

8. Abstract, lines 17-18, p1: Specify analytical methods in “Methods: in vitro digested CBS and CBS-enriched ice cream component characterization and antioxidant capacity were analyzed.”

9. Introduction, lines 39-41, p1: Add references to support “Cocoa beans, the seeds of the tropical Theobroma cacao L. tree, are the source of the main ingredient used for chocolate and cocoa-derivative food production, namely cocoa mass (liquor), a solid or semi-solid paste containing both cocoa powder and cocoa butter in roughly equal proportion.”

10. Introduction, lines 54-56, p2: Add references to support “Polyphenols are secondary plant metabolites, chemically characterized by the presence of several phenolic rings with two or more hydroxyl groups that make them real free radical scavengers and metal chelators.”

11. Introduction, lines 59-60, p2: Add references to support “Intestinal mucosa is particularly exposed to environmental and dietary agents responsible for sustaining immune and inflammatory responses.”

12. Introduction, lines 63-65,56, p2: Add references to support “The dietary animal fats typically present in the Western diet, as well as the lipid oxidation products arising from their processing, have been suggested to exert inflammatory and oxidative insults in gut mucosa.”

13. Introduction, lines 69-72, p2: Add references, specify MAPkinase and NF-kB n “We have previously proven that a mixture of oxysterols corresponding to high dietary consumption of cholesterol triggered oxidative reactions and inflammatory responses in human enterocyte-like Caco-2 cells. NADPH oxidase, MAPkinase and NF-kB appeared to be the main signaling pathways involved.”

14. Introduction, lines 73-74, p2: Add references to support “The activation of antioxidant defense in response to cell oxidative conditions is common knowledge”

15. Introduction, lines 78-79, p2: Add references and specify Nrf2 in “Nrf2 is a key transcription factor whose activation represents a protecting adaptive cell response against redox stressors by  promoting codification for various stress-related detoxifying and antioxidant enzymes”

16. Introduction, lines 81-83, p2: Add references “Recent studies showed that dietary oxysterols were able to induce inflammation in CaCo-2 cells by activating immune system-related Toll-like Receptor (TLR) superfamily members TLR2 and TLR4.”

17. Materials, line 94, p2: Specify HPLC-DAD in “Chemicals for HPLC-DAD analyses:”

18.Materials, lines 113-115, p3: Specify DTT in “Goat anti-rabbit IgG-Alexa Fluor 488, goat anti-mouse IgG-Alexa Flour 488, Lithium dodecyl sulfate (LDS) Sample Buffer 4X and DTT Sample Reducer 10X were purchased from Thermo Fisher Scientific (Life Technologies Italia, Monza, Italy).”

19. Preparation of CBS samples, Title, line 128, p3: Replace CBS by its full name in the title.

20. Preparation of CBS samples, lines 128-130, p3: Specify composition and purity of cocoa bean shell to support “Pastiglie Leone Srl (Turin, Italy) kindly supplied cocoa bean shell (CBS) samples yielded from cocoa beans of Forastero cultivar from Sao Tomé.”

21. Preparation of CBS samples, lines 134-137, p3: Specify composition of ice cream to support “CBS samples were then used as fat replacers in the preparation of ice cream (IC) at different concentrations as previously described by Barbosa-Pereira and colleagues [23]. The ice cream obtained by 4% fortified CBS flour (CBS-IC) was selected because of its reduced fat and increased fiber contents compared with plain IC (considered as control sample).”

22. Preparation of CBS samples, lines 139-141, p3: Indicate percentage range of CBS in “The percentage of CBS powder used in IC was chosen after food tasting, and CBS- enriched IC was produced by maintaining structural characteristics like plain chocolate ice cream.”

23. In vitro simulated gastrointestinal digestion line 145 p3 to line p4 150: Add more information to support “The simulated gastrointestinal digestion related to the oral, gastric, and small intestinal tracts was performed according to the routine standardized static in vitro method suitable for food described by Minekus and collaborators [24]. For each digestion phase, digestive juices were prepared for mouth (Simulated Saliva Fluid, SSF), stomach (Simulated Gastric Fluid, SGF) and small intestine (Simulated Duodenal Fluid, SDF) following protocol indications as described by the authors.”

24. Reverse phase – high performance liquid chromatography (RP-HPLC-PDA) analysis of CBS and CBS-ICcomponents, Title, line 167-168, p4 : “Delete (RP-HPLC-PDA), replace CBS by its full name in the title:

25. Reverse phase – high performance liquid chromatography (RP-HPLC-PDA) analysis of CBS and CBS-ICcomponents, Title, line 173-175 p4 : Specify bioactive compounds in “The bioactive compounds were separated at 35 °C on a reverse phase Kinetex Phenyl-Hexyl C18 column (150 × 4.6 mm internal diameter and 5 μm particle size) (Phenomenex, Castel Maggiore, Italy).”

26. Cell culture and treatments, lines 189-194, p4: Justify the use of cells and add references to support “The Cell Bank Interlab Cell Line Collection (Genoa, Italy) provided human colorectal adenocarcinoma CaCo-2 cells (accession number: ICLC HTL97023). Cells were plated at 1×106/ml density, cultured in DMEM supplemented with 10% heat inactivated FBS, 1% antibiotic/antimycotic solution (100 U/ml penicillin, 0.1 mg/ml streptomycin, 250 ng/ml amphotericin B and 0.04 mg/ml gentamicin) and maintained at 37°C in a humidified atmosphere containing 5% CO2.”

27. Cell culture and treatments, lines 194-196, p4: Add information how cell phenotype was checked to support “To allow their spontaneous differentiation into enterocyte-like phenotype, cells were grown for additional 18 days after reaching confluence.”

28. Cell culture and treatments, lines 197-198, p4: It was unclear if the cells were incubated for 18 days before each treatment in “Before each treatment, CaCo-2 cells were incubated in serum-free medium overnight to make them quiescent.”

29. Cell culture and treatments lines 201-205 , p5: Specify 7K in “The percentage composition of oxysterols used in the Oxy-mix was 42.96% for 7K, 32.3% for α-epox, 5.76% for β-epox, 4.26% for 7α-HC, and 14.71% for 7β-HC. The molarity of each oxysterol in 60 μM Oxy-mix was calculated as 25.8 μM 7K, 19.4 μM α-epox, 3.4 2μM β-epox, 2.6 μM 7α-HC, 8.8 μM 7β-HC by considering an average molecular weight of 403 g/mol [20].”

30. Cell culture and treatments, lines 206-208, p5: It was unclear if the cells were subjected to a 18-day incubation prior to the treatment in “Cells were pre-incubated with CBS or CBS-IC extracts for 1 h before the Oxy-mix treatment. Based on cell death analyses, CBS or CBS-IC were used to reach final 5% concentration in the cell culture.”

31. Cell death evaluation, lines 213-215, p5: Specify types of volume v/v in “The extracellular release of lactate dehydrogenase (LDH) was considered as a parameter of cell death. Cells were treated with increasing concentrations [5%, 10%, 30%, 50% (v/v)] IC, CBS-IC or CBS, and incubated or not with 60 μM Oxy-mix.”

32. Cell death evaluation, lines 217-221, p5: Specify number of independent measurements to support statistics in “LDH was evaluated spectrophotometrically at 340 nm wavelength by recording NADH production/min. LDH release was expressed as a percentage of the total enzyme released into cell culture medium by complete cell lysis (obtained by 0.5% Triton X-100 addition to the plate containing the same cell density as the treated cells) (Table S1 supplementary materials).”

33. Total Phenolic, Tannin and Flavonoid Contents, lines 223-224, p5: Add more information to support “The amount of total phenolics (TPC), flavonoids (TFC) and tannins (TTC) in CBS and CBS-IC was quantified as reported by Barbosa-Pereira and colleagues [23].”

34. Total Phenolic, Tannin and Flavonoid Contents, lines 226-227, p5: Independent experiments shall be performed instead of triplicates to support statistics in “All the analyses were performed in triplicate.”

35. Antioxidant Capacity, lines 235-238, p5: Indicate number of independent measurements to support statistics in “The inhibition percentage (IP) of DPPH radical was calculated by using the following equation: IP (%) = [(A0 - A30)/A0] x 100 (where A0 is the absorbance at time 0, and A30 the absorbance after 30 min). A standard curve of trolox was used (12.5–300 μM) to assess the radical-scavenging activity values, which were expressed as μmoles of trolox equivalents (TE) for ml of sample (μmol TE/ml).”

36. Evaluation of IL-8 and MCP-1 protein levels by ELISA, Title line 240, p5: Replace IL-8, MCP-1 and ELISA by their full names in the title.

37. Evaluation of IL-8 and MCP-1 protein levels by ELISA, lines 242-243, p5: Specify if cells were fully differentiated in “Controls were referred to untreated cells incubated in 1% FBS DMEM alone for the same time as treated cells.”

38. Evaluation of IL-8 and MCP-1 protein levels by ELISA, lines 243-244, p5: Add more information how ELISA was performed to support “Cytokine concentrations were analyzed and estimated by using commercial ELISA kits according to the manufacturer's instructions.”

39. Evaluation of IL-8 and MCP-1 protein levels by ELISA, lines 246-247, p5: Replace optical density by absorbance in “Optical density at 655 nm wavelength was considered as a value reference for each sample.”

40. Evaluation of IL-8 and MCP-1 protein levels by ELISA, lines 251-252, p6: Independent experiments shall be performed instead of triplicates to support statistics in “The analyses were performed in triplicate, data calculated by using SlideWrite Plus software (Advanced Graphics Software, Rancho Santa Fe, CA, USA) and values expressed as pg cytokines/mg cell culture medium proteins.”

41. Tight Junction protein immunoblotting, lines 255-256, p6: Specify if cells were fully differentiated and PBS buffer composition in “Cells were scraped and washed with ice-cold phosphate buffer saline (PBS) at the end of each treatment.”

42. Tight Junction protein immunoblotting, lines 256-259, p6: Replace (w/v) by units in “One-hundred and fifty μl lysis buffer were added for protein extraction [PBS supplemented with 1% Triton X-100 (v/v), 1% sodium dodecyl sulfate (w/v) (final volume)]. Once lysed, samples were incubated for 30 min on ice and centrifuged at 12,052 x g at 4 °C for 15 min.

43. Tight Junction protein immunoblotting, lines 258-259, p6: Add more information to support “Total cell extract protein concentration was evaluated with Bio-Rad protein assay dye reagent.”

44. Tight Junction protein immunoblotting, lines 264-267 and lines 269-273, p6: Replace (w/v) by units in “The membranes were then incubated at room temperature for 1 h in TBS supplemented with 0.05% (v/v) Tween 20 [TTBS] blocking buffer plus 5% (w/v) skimmed milk powder (final volume).” And in “Three sub- sequent washes in TTBS were performed and blots were incubated with anti-rabbit or anti-mouse HRP-conjugated IgG (1:1000 dilutions) in TBS with 0.1% Tween-20 (v/v) and 5% skimmed milk powder (w/v) for 1 hour. Finally, blots were washed twice in TTBS for 10 min.”

45. Tight Junction protein immunoblotting, lines 275-276, p6: and Indicate number of independent measurements to support statistics in “Protein band den-sities were quantified by using Image J Software (Bethesda, Maryland, USA).”

46. Tight Junction protein immunofluorescence, lines 278-280, p6: Specify how cells were monitored to indicate their differentiation state in “Cell localization of TJ proteins (claudin 1, occludin and JAM-A) was visualized by using immunofluorescence technique. For this analysis, treatments were performed by using differentiated CaCo-2 cell monolayers grown on 13 mm diameter glass slides.”

47. Real-time quantitative reverse-transcription polymerase chain reaction (qRT-PCR), Tittle, line 299, p7: Delete (qRT-PCR) in the title.

48. Real-time quantitative reverse-transcription polymerase chain reaction (qRT-PCR), Tittle, lines 300-301, p7: Specify if cells were differentiated in“Cells were pre-treated or not with CBS, CBS-IC for 1h and incubated with Oxy-mix for 6 h.”

49. Real-time quantitative reverse-transcription polymerase chain reaction (qRT-PCR), line 305, p7: Add more information how total m-RNA was extracted in “Total mRNA was extracted from treated cells using TRIzol™.”

50. Real-time quantitative reverse-transcription polymerase chain reaction (qRT-PCR), lines 309-311, p7: Specify qRT-PCR here and not in the title in “qRT-PCR was performed on 20 ng cDNA using TaqMan gene expression probes for Nrf2, and amplified cDNAs were analyzed by 7500 Fast real-time PCR system (Applied Biosystems, Thermofisher, Monza, Italy).”

51. Statistical analyses, lines 323-324, p7: Specify number of independent samples to substantiate statistics in “All values were expressed as mean ± Standard Deviation (SD) and data were analyzed with GraphPad InStat software (San Diego, CA, USA).”

52. CBS-enriched ice cream shows high recovery of phenolic and methylxanthine compounds, and antioxidant capacity, Title, line 326, p7: Replace CBS by its full name in the title.

53. CBS-enriched ice cream shows high recovery of phenolic and methylxanthine compounds, and antioxidant capacity, line 328-330, p7: The comparisons between IC and CBS-IC that is more relevant than that between CBS-IC and CBS for consumer acceptance since apparently IC is devoid of CBS. Rephrase “Chromatographic analyses of CBS and ice cream showed that CBS-IC maintains similar structural characteristics to the common chocolate ice cream, making it suitable for consumer acceptance [8].”

54. CBS-enriched ice cream shows high recovery of phenolic and methylxanthine compounds, and antioxidant capacity, Table 1 line 339-341, p7: Specify Sig between which pairs in “Data are expressed as mean values (n=3) ± standard deviation. Sig: significance. ANOVA associated Duncan’s test was used as a multiple range test to highlight the significant differences among all the treatments. ANOVA significance: ***p 3<0.001. Means showing different lowercase letters within the same line are significantly different at p <0.05.”

55. CBS-enriched ice cream shows high recovery of phenolic and methylxanthine compounds, and antioxidant capacity, line 339-341, p7: Specify if there were independent measurements and not triplicate to substantiate “Data are expressed as mean values (n=3) ± standard deviation. Sig: significance. ANOVA associated Duncan’s test was used as a multiple range test to highlight the significant differences among all the treatments. ANOVA significance: ***p <0.001. Means showing different lowercase letters within the same line are significantly different at p <0.05.”

56. CBS-enriched ice cream shows high recovery of phenolic and methylxanthine compounds, and antioxidant capacity,Table 2, line 344-349, p7: Specify Sig between which pairs in “Total phenolic content (TPC), total flavonoid content (TFC), total tannin content (TTC) and antioxidant capacity (RSA) were evaluated in plain ice cream (IC), 4% CBS-fortified ice cream (CBS-IC) and CBS powder (CBS) obtained after gastro-intestinal digestion. TPC are expressed as gallic acid equivalent (GAE); TFC and TTC as catechin equivalent (CE); RSA as trolox equivalent (TE)/ml extracts. Data are expressed as mean values (n=3) ± standard deviation. Sig: significance. ANOVA associated Duncan’s test was used as a multiple range test to highlight the significant differences among all the treatments. ANOVA significance: ***p <0.001. Means showing different lowercase letters within the same line are signifi-cantly different at p <0.05.:”

57. CBS-enriched ice cream shows high recovery of phenolic and methylxanthine compounds, and antioxidant capacity,Table 2, line 344-349, p8: Specify if there were independent measurements and not triplicate to substantiate “Total phenolic content (TPC), total flavonoid content (TFC), total tannin content (TTC) and antioxidant capacity (RSA) were evaluated in plain ice cream (IC), 4% CBS-fortified ice cream (CBS-IC) and CBS powder (CBS) obtained after gastro-intestinal digestion. TPC are expressed as gallic acid equivalent (GAE); TFC and TTC as catechin equivalent (CE); RSA as trolox equivalent (TE)/ml extracts. Data are expressed as mean values (n=3) ± standard deviation. Sig: significance. ANOVA associated Duncan’s test was used as a multiple range test to highlight the significant differences among all the treatments. ANOVA significance: ***p <0.001. Means showing different lowercase letters within the same line are signifi-cantly different at p <0.05.:”

58. CBS and CBS-enriched ice cream prevent IL-8 and MCP-1 production from differentiated CaCo-2 cells treated with pro-inflammatory Oxy-mix, Title lines 355-356, p8: Replace CBS, IL-8, MCP-1 and Oxy-mix by their respective full names in the title.

59. CBS and CBS-enriched ice cream prevent IL-8 and MCP-1 production from differentiated CaCo-2 cells treated with pro-inflammatory Oxy-mix, lines 358-361, p8: Replace (w/v) by units in “Based on the cytotoxicity analyzed as LDH release, 5% (w/v) CBS-IC/CBS (cell culture final concentration) showed no cytotoxic effects both in presence and absence of the Oxy-mix and was therefore chosen as the best concentration for cell treatment.”

60. CBS and CBS-enriched ice cream prevent IL-8 and MCP-1 production from differentiated CaCo-2 cells treated with pro-inflammatory Oxy-mix, lines 363-366, p8: Indicate Figure or findings to support “CBS-IC and CBS anti-inflammatory efficacy was evaluated in terms of IL-8 and MCP- 1 production. As expected, Oxy-mix induced a strong increase in the pro-inflammatory cytokines released by CaCo-2 cells in the culture medium (2.25- and 3.3-fold increase of IL-8 and MCP-1, respectively).”

61. Tight junction impairment induced by the Oxy-mix is prevented in differentiated CaCo-2 cells pre-treated with CBS-IC or CBS, Title lines 384-385, p9: Replace Oxy-mix, CBS-IC and CBS by their respective full names in the title.

62. Tight junction impairment induced by the Oxy-mix is prevented in differentiated CaCo-2 cells pre-treated with CBS-IC or CBS, lines 386-389, p9: It was unclear how cell differentiation was monitored to support “To elucidate the potential beneficial effect of CBS formulations in maintaining the intestinal epithelial barrier integrity, we examined protein levels and cellular distribution of the main tight junctions (TJs) in differentiated CaCo-2 cells cultured in monolayers.”

63. Tight junction impairment induced by the Oxy-mix is prevented in differentiated CaCo-2 cells pre-treated with CBS-IC or CBS, lines 415-417, p11: in vitro shall be in italic in “The antioxidant capacity observed for the CBS formulations led us exploring the in vitro potential effect of these extracts in modulating the cell adaptive response other than their direct scavenging activity.”

64. Tight junction impairment induced by the Oxy-mix is prevented in differentiated CaCo-2 cells pre-treated with CBS-IC or CBS, Title lines 430-431, p12: Specify if there were independent measurements in “Data are reported as means ± SD of three experiments per-formed in triplicate.”

65. Theobromine can preserve oxysterol-mediated tight junction derangement and inflammation but does not reinforce antioxidant cell response, lines 438-439, p12: It was unclear what was the differentiation state of the cells in “Therefore, CaCo-2 cell monolayers were pre-treated with 13 μM theobromine for 1 h and then treated with the Oxy-mix.”

66. Discussion, lines 467-468, p13: Merge into introduction “The perturbation of the intestinal barrier structure and function is an important fea-ture of gut inflammation; this process playing a critical role in the pathogenesis of human 467intestinal disorders such as inflammatory bowel diseases [27].”

67. Discussion, lines 468-472, p13: Add references to support “Intestinal mucosal damage results from tight junction loss, increased paracellular transport, apoptosis, and transcel-lular permeability. A leaky gut may be a crucial event allowing digestive metabolites and bacteria-derived molecules to enter the mucosa and trigger inflammatory processes, thus inducing/amplifying the barrier damage.”

68. Discussion, lines 474-476, p13: Merge into introduction “The Western style diet represents one of the main risks for altered intestinal integrity [28]. On the contrary, a dietary regimen rich in fruit and vegetables and/or the intake of supplements able to reduce inflammation may help to maintain intestinal barrier integrity by reducing development or at least preventing disease relapse.”

69. Discussion, lines 500-502, p14: Add references to support “The cocoa bean shell arising from the remnants of the chocolate production process has recently attracted attention as a potential raw material to be employed in food supple- ments and functional beverages.”

70. Discussion, lines 506-510, p14: So far bacterial damages were not the object of this work. Rephrase “The production of these two chemokines in-creases during intestinal mucosal damage caused by bacteria toxin exposure [36]. Consist-ently, both CBS extracts restored protein levels of the main TJs involved in the regulation of paracellular permeability claudin 1, occludin and JAM-A, which were altered in oxys- terol-treated CaCo-2 cells.”

71. Discussion, lines 525-527 p14: Add explanations and references to supportCBS beneficial property in preserving intestinal cell layer from permeability altera-tion could be due to the high concentration of antioxidant flavonoids, in particular cate-chins.

72. Discussion, lines 561-563, p15: Due to the lack of findings on flavonoids and inconclusive findings on theobromine delete “These data confirm our hypothesis that CBS protective action could be due to the interaction of various components - including flavonoids, and partially theobromine - which together contribute in giving the biological beneficial effects of this by-product.”

73. Conclusion, lines 568-570, p15: Add references to support “Various studies sug-gest cocoa bean shell reuse in functional food because of its richness in polyphenols’ po-tential benefit on health.”

Reviewer 3 Report

Authors should also mention; what is the difference between palatability for CBS enriched ice cream vs normal ice cream?

High Catechin in CBS containing ice cream might be good as antioxidant but it may also have anti-nutrient properties. Will high Caffeine make this ice cream more addictive?

It would have been wise to use macrophage cells for Oxy-mix along with CBS.

Why data in Figure 1 expressed in pg/mg of protein. It should be pg/ml. I assume most of ELISA is performed on supernatant not on the whole cell extract (WCE). ELISA is for secretory protein, WCE is misleading (you are missing the secretory mature bioactive cytokines in WCE). Please redo the experiment on supernatant. 

Figure 2 Statistics (###) does not look significantly different. Would you like to see the statistics again carefully? 4th bar (CBS) is tilted towards the X axis. I think it is a software glitch. Was it in N=3 samples? If not; please repeat for N=3.

Please put Nrf2 western. RT PCR changes close to 1.5-fold are not very significant to be considered worth reporting. When you have so much of standard deviation it means it could be anything…...over here again the statistics is uncertain and unpredictable.

Figure 5: please use supernatant for ELISA. For 5.D please use western.

Round 2

Reviewer 1 Report

The manuscript has been significantly improved, thus I am recommending it to further processing.

Author Response

The Authors sincerely thank the reviewer for encouraging suggestions and favorable comments.

Reviewer 2 Report

General comments :

Three general comments 3, 5 and 7 were insufficiently addressed.

3- It was unclear why human colorectal adenocarcinoma CaCo-2 cell line was used. There was no characterization of the cells and it was unclear how cell differentiation, if any, was monitored.

R.: This type of cells has the peculiarity to spontaneously differentiate as intestinal cells having the absorption function typical of normal enterocytes. This model is well known for these features in the scientific community, and therefore it is an excellent model for mimicking intestinal mucosa. Cell differentiation is routinely monitored by alkaline phosphatase in our laboratory. Following the reviewer suggestion, more information and references have been added accordingly (see lines 244-253, Refs. 37, 38 in the revised manuscript)

Answer : There was still insufficient information how cell differentiation was monitored. It was unclear if cell differentiation was carefully checked. The only information as mentionned in lines 255-256 « CaCo-2 differentiation grade was routinely performed by monitoring alkaline phosphatase expression [38]. » was insufficient since there were no values of alkaline phosphatase activity. Other cell markers shall be determined, since there are many cells that express tissue non specific alkaline phosphatase.. At this stage , there was insufficient cell characterization.

5- The overall findings appear preliminary due to the lack of assays on other different cell lines, or primary cells, inconclusive findings on the effects of theobromine as well as insufficient information on the bioactive compounds. Flavonoids were suspected to be beneficial but there were no experiments to assess their eventual beneficial effects.

R.: As underlined above, these cells well mimic mucosal intestinal monolayer with normal permeability function (please see the so wide scientific literature about it). The aim of the present investigation is to evaluate these compounds for their properties in preventing layer damage. Most of the available studies report the beneficial effects of single flavonoids, but very, very few data focus on the CBS biological impact, as well as on the possible theobromine contribution to inflammation and permeability preservation. Nevertheless, we agree with the Reviewer that, beside theobromine, we need to evaluate also the individual effects of other molecules present in CBS (and also to deepen theobromine biological value). These analyses are in progress and will be the object (we hope) for further publications.

Answer : What was missing is the effects of threobromine on other cell lines or on primary cells. Therefore the overall findings were quite preliminary since they originated from one single cell line, poorely characterized. At least, several distinct cell lines shall be tested, combined whenever possible with primary cells to obtain reliable findings, otherwise the findings remain preliminary.

7 ; There were too much self-citations (10 over 46). References N° 8,16, 17,18,19,20,22,23,32 and 46.

R.: The references related to CBS are previous findings from our research group and we think that they are necessary to explain the results and give the readers a state of art for which some present authors have a main contribution. However, Ref. 23 of the original manuscript has been removed

Answer : It was not addressed, the authors added three more references, even including one deleted. There were 13 self-citation. References 9, 20, 23, 27, 28, 29, 30, 32, 33, 35, 48, 52 and 65, confirming the incremental nature of the findings.

Reviewer 3 Report

Authors response to reviewer is acceptable.

Current article is improved significantly then its previous versions. Only thing I did not feel convinced is statistics. Anyway, overall paper is in good shape.

Best,

Author Response

(The authors gave the same response as above.)
